# High-throughput FastCloning technology: A low-cost method for parallel cloning

**Hua Jiang[1]☯, Fan Meng[2]☯, Deren Lu[1], Yanjuan Chen[1], Guilin Luo[1], Yuejun Chen[1], Jun Chen[2], Cheng Chen[3], Xi Zhang[4], Dan Su[1,5]***

**1** State Key Laboratory of Biotherapy and Cancer Center, West China Hospital, Sichuan University and Collaborative Innovation Center of Biotherapy, Chengdu, Sichuan, P.R. China, **2** Jujing-Chengdu Biotech, Chengdu, Sichuan, P.R. China, **3** Department of Gynecology and Obstetrics, Chongqing General Hospital, University of Chinese Academy of Sciences, Chongqing, P.R. China, **4** School of Mechatronic Engineering and Automation, Shanghai University, Shanghai, P.R. China, **5** Tianjin International Joint Academy of Biotechnology and Medicine, Tianjin, P.R. China

☯ These authors contributed equally to this work.
\* sudan@scu.edu.cn

**Data Availability Statement:** All relevant data are within the paper and its Supporting Information files.

**Funding:** This work was supported by grants from the National Key Research and Development Program of China (2017YFA0505903), the National

## Abstract

FastCloning, a reliable cloning technique for plasmid construction, is a widely used protocol in biomedical research laboratories. Only two-step molecular manipulations are required to add a gene (cDNA) of interest into the desired vector. However, parallel cloning of the gene into multiple vectors is still a labor-intensive operation, which requires a range of primers for different vectors in high-throughput cloning projects. The situation could even be worse if multiple fragments of DNA are required to be added into one plasmid. Here, we describe a high-throughput FastCloning (HTFC) method, a protocol for parallel cloning by adding an adaptor sequence into all vectors. The target gene and vectors were PCR amplified separately to obtain the insert product and linear vectors with 18-base overlapping at each end of the DNAs required for FastCloning. Furthermore, a method for generating polycistronic bacterial constructs based on the same strategy as that used for HTFC was developed. Thus, the HTFC technique is a simple, effective, reliable, and low-cost tool for parallel cloning.

## Introduction

Plasmids are small circular DNA molecules within cells that are physically isolated from chromosomal DNA and can replicate independently [1]. Plasmids are valuable tools in the field of genetic engineering and are widely used in recombinant cloning technology to transfer foreign genetic material into different cell types. Plasmid cloning is a major procedural cornerstone in the fields of molecular biology and biochemistry, owing to its extensive use in various applications, such as gene cloning, transfection, RNA interference, recombinant protein expression, gene therapy, and DNA manipulation [2–5]. Plasmid cloning is no longer limited to type II restriction enzymes, although this method has been routinely used in many research groups since the insertion of *Xenopus laevis* rDNA fragments into the Psc101 plasmid was first reported in 1974 [6]. The traditional "cut and paste" cloning procedure requires one or two types of restriction enzyme to cut both target DNA and plasmids with blunt/sticky ends, in

Natural Science Foundation of China (31370735, 31670737 to D. S., 81671494 to C. C.), the Technology Department of Tianjin Foundation (19YFZCSN00470), 1·3·5 projects for disciplines of excellence, West China Hospital, Sichuan University (ZYJC18033), the Special Research Fund on COVID-19 of Sichuan Province (2020YFS0010), the Key Project on COVID-19 of West China Hospital, Sichuan University (HX-2019-nCoV-044), and the Natural Science Foundation of Sichuan Province(2022NSFSC0008).

**Competing interests:** The authors have declared that no competing interests exist.

**Abbreviations:** HTFC, high-throughput fast cloning; LIC, ligation-independent cloning; SLIC, sequence ligation-independent cloning; PIPE, polymerase incomplete primer extension; IPTG, isopropyl β-ᴅ-1-thiogalactopyranoside; TEV, tobacco etch virus; RBS, ribosomal binding site.

conjunction with ligation using *Escherichia coli* DNA ligase or bacteriophage T4 DNA ligase. Over the past few decades, considerable efforts have been made to simplify and standardize cloning processes, allowing a vast array of target DNAs to be more easily assembled in plasmids. Recently, several techniques, such as T. A. cloning [7], Gateway [8], plasmid fusion system [9], Golden Gate [10], sequence and ligation-independent cloning (LIC) [11], and FastCloning [12] have been developed to enhance cloning efficiency, reduce costs, and minimize process times.

Each developed cloning method has its own associated advantages and disadvantages. T. A. cloning simplifies traditional restriction and ligation cloning with a one-step cloning strategy using *Taq* DNA polymerase, which attaches a single deoxyadenosine to the 3' end of a PCR product. Linearized vectors with complementary 3′-deoxythymidine residues enable efficient ligation of the insert into a vector [13]. The Gateway system requires two enzymes (BP Clonase and LR Clonase) to transfer the target DNA fragment between plasmids bearing compatible flanking recombination attachment sites. However, the DNA fragment of interest must be inserted into the entry vector using a proprietary set of recombination sequences and donor vector before selecting the most appropriate destination vector for the experimental requirements [8, 14, 15]. The Golden Gate cloning method relies on type IIS restriction enzymes (*Bsa*I and *Bpi*I/*Bbs*I), as these cleave outside their recognition sequences, first discovered in 1996. Given that these overhangs are not part of the recognition sequence, they can be used for direct assembly of DNA fragments. However, the Golden Gate cloning method is not 100% sequence-independent [10, 16]. Similar to the Golden Gate approach, uracil-specific excision reaction (USER^TM) technology adopts a strategy based on the use of two enzymes, uracil DNA glycosylase and DNA glycosylase-lyase endonuclease VIII, to generate DNA fragments flanked by 10-base overhangs from PCR reactions using primers containing 2′-deoxyuridine as a substitute for 2′-deoxythymidine [17, 18]. In 2009, Gibson et al. developed a novel method for the ready assembly of multiple linear DNA fragments [19], in which, regardless of fragment length or end compatibility, multiple overlapping DNA fragments could be connected in a single isothermal reaction using three different enzymes. However, these methods require the presence of appropriate restriction enzymes and modifying enzymes for manipulation, making the procedures costly, especially in high-throughput experiments.

To avoid the use of restriction endonucleases and DNA ligase, a cloning procedure called ligation-independent cloning (LIC) was developed, which requires PCR products and linearized vectors flanked by complementary sequences of 10–18 bp [20]. Single-stranded DNA overhangs are generated by incubating the PCR fragment and linearized vector with T4 polymerase in the presence of only a particular nucleotide that is absent in the overhang sequence [21]. Although LIC is a reliable cloning method, the technique is limited by the 10–18 base overhangs that lack one of the four nucleotides present in the reaction, thereby preventing T4 polymerase from digesting the entire 10 to 18 bases. In 2007, Elledge et al. refined the LIC procedure to develop sequence ligation-independent cloning (SLIC) [22], which can generate imperfect recombination intermediates via PCR and imprecise T4 exonuclease activity, thereby overcoming the requirement for carefully designed DNA overhangs used in LIC. Next, several ligation-independent cloning methods have also been developed, including polymerase incomplete primer extension (PIPE) cloning and overlap extension cloning (OEC) [23, 24]. These strategies are dependent on the generation of complementary overhangs by DNA polymerase, without the requirement for specific restriction enzymes or ligases. After a decade of technical upgradation, cloning methods can be classified into four groups: traditional restriction endonuclease, ligation-dependent, *in vitro* assembly, and *in vivo* assembly methods (Fig 1).

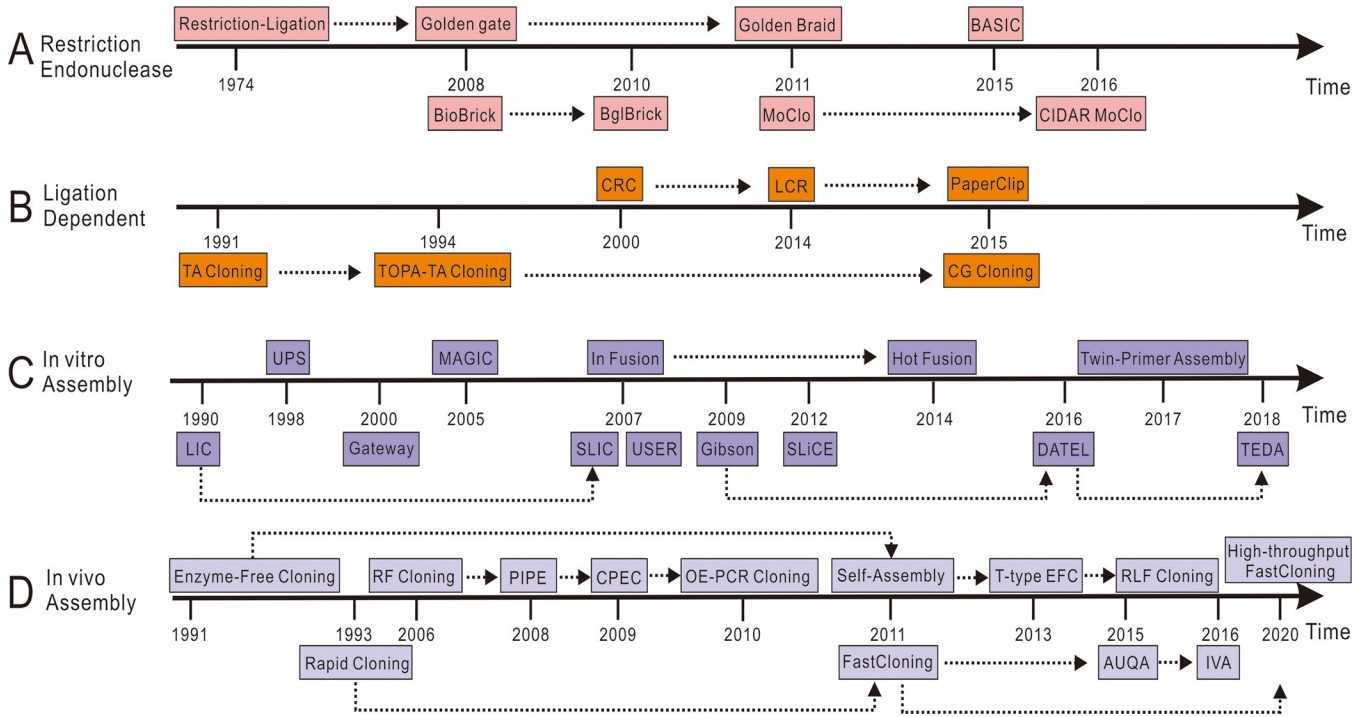

**Fig 1. A timeline of molecular cloning techniques.** The available cloning techniques can be classified into four types: (A) restriction endonucleases, (B) ligation-dependent staining, (C) *in vitro* assembly, (D) *in vivo* assembly.

Based on this technical development, an extensive range of commercial cloning kits using LIC or SLIC technology are currently available on the market, including Gibson Assembly (NEB), In-Fusion (Takara), GeneArt Seamless Cloning (Thermo Fisher Scientific), Fast Seamless Cloning (Dogene), Gold Fusion (SBI), and CloneEZ (Genscript). Although these commercial kits can assist researchers in the design and construction of more efficient recombinant plasmids, they tend to be comparatively expensive. Numerous protocols exploit the recombination systems of λ or P1 bacteriophages and bacteria to enhance the efficiency of recombinant plasmid cloning, including Gateway, vector plasmid-fusion system (UPS) [9, 25], mating-assisted genetically integrated cloning (MAGIC) [25], FastCloning [12] and seamless ligation cloning extract (SLiCE) technologies [26]. Nevertheless, although the efficiency of recombinant plasmid cloning has been enhanced by adopting alternative cloning strategies, each of these techniques has its limitations, such as complex protocols, time-consuming procedures, expensive reagents/equipment, and the need for specific expertise, notably in the case of high-throughput plasmid construction [27]. With the advancement in omics and big data, functional studies are pressing the demand for a viable high-throughput multiple-vector cloning method. Owing to its simplicity and efficiency, FastCloning has highlighted its potential role and importance in high-throughput plasmid construction. However, first-generation FastCloning is still dependent on endonuclease *Dpn*I digestion to eliminate background template vectors and does not facilitate high-throughput construction that enables cloning of a single gene in parallel vectors, which is of great need in recombinant protein engineering.

Owing to its ease of manipulation, rapid growth, and convenient low-cost handling, *E. coli* remains the most extensively employed expression and co-expression host [28]. Moreover, the application of affinity tags is of particular importance with respect to the harvesting and purification of recombinant proteins from host cell extracts, as well as for enhancing protein

solubility [29]. To date, versatile affinity tags have been categorized as histidine tags used in conjunction with immobilized metal affinity chromatography, epitope tags used in immunoaffinity isolation and detection, ligand-binding affinity tags, and other expression chaperone tags [30]. All prokaryotic expression vectors are characterized by specific features, such as unique promoters, affinity or fusion-promoting tags, and location, which significantly affect the expression, solubility, and stability of recombinant proteins. Consequently, high-throughput construction is the initial step in the screening for these associated factors. Recently, single-particle cryo-electron microscopy (cryo-EM) has become an increasingly widely used technique employed by structural biologists to elucidate large protein complexes that require the production of pure, stable, and homogenous samples [31]. Co-expression is a dominant strategy based on the *in vivo* assembly of individual protein subunits following heterologous expression of multi-gene constructs, which can be achieved by adopting alternative strategies [32]. Given their ease of manipulation and wide compatibility, polycistronic expression systems are widely used for the co-expression of multiple protein complexes, such as the pST39 or pST44 system [33–35]. However, each strategy has been developed using different recombinant DNA cloning methods, and requires specialized vectors from diverse sources.

To circumvent these problems, we developed a feasible and versatile DNA assembly method called high-throughput FastCloning (HTFC), which does not require any restriction endonucleases or entry plasmids for the final recombination process in cells. Using this method, we designed an insert sequence comprising a *ccdB* gene expression cassette flanked by two 18-bp adapter sequences (recognized by HighR and HighF primers), which was inserted into the 12 prokaryotic expression vectors used in this study. All vectors and target cDNA were linearized and amplified by PCR using high-fidelity DNA polymerase. The same adapter sequences (HighR and HighF) were added to all linear vectors and the target cDNA during the PCR cycles. The unpurified PCR products of cDNA were mixed with diverse linear vectors. Finally, all mixtures were simultaneously transformed into competent cells to obtain target clones. In the process of selecting positive clones, we used the gene replacement principle in combination with negative *ccdB* selection without the need for *DpnI* pre-treatment for the removal of template DNA plasmids. Using the prepared linearized vectors, a single insert fragment could be simultaneously assembled into 12 vectors with high efficiency. Furthermore, we describe a conventional method for generating polycistronic expression systems based on the HTFC strategy. This method was designed to simultaneously prepare various polycistronic expression systems using combinations of expression cassettes. Thus, this method of HTFC can be widely used for simultaneous cloning, not only for the single cDNA of a gene into an array of different expression vectors, but also for multiple expression cassettes into one vector with different combinations.

## Materials and methods

### Design of high-throughput prokaryotic expression vectors

Twelve versatile prokaryotic expression vectors were modified to generate first-generation high-throughput prokaryotic expression vectors by adding adapter sequences to the original vectors, which have been widely used to express recombinant proteins attached to different affinity tags including 6×His (histidine), 6×His-SUMO (small ubiquitin-like modifier), GST (glutathione-*S*-transferase), and 6×His-MBP (maltose binding protein) tags. The details of these vectors are listed in **S1 Table**. Using the FastCloning technique [12], we first inserted a 36-bp dsDNA sequence 5′-CTGGTGCCGCGCGGCAGCGGAGGAGGAATCATCATC-3′ as an adapter into pSDV1-V12 vectors, which included a thrombin site to enable the removal of an N-terminal affinity tag. For the insertion of the 36-bp sequence, one of two strategies can be

adopted, depending on whether the original vector contains a thrombin site (shown in S1 Fig).
To efficiently excise the C-terminal affinity tag, a TEV site (GAAAACCTGTATTTTCAGGGC)
was inserted downstream of the 36-bp marker element. All primers used in these procedures
are listed in **S2 Table**.

   To eliminate the necessity for *Dpn*I in digesting the template circular plasmid, a lethal gene
(*ccdB)* expression cassette was introduced into the pSDV1–V12 vectors in the middle of a
36-bp adapter, designated pSDB1 to pSDB12. The *ccdB* expression cassette includes a *cat* pro-
moter, chloramphenicol-resistance gene, and *ccdB* gene, which would normally facilitate the
expression of the lethal *ccdB* protein in ubiquitous *E. coli* cloning strains, such as DH5α,
JM109, and DH10B. Recombinant plasmids containing *ccdB* were used as templates to linear-
ize the vectors.

## Recommended procedure for HTFC

First, the primers for efficient amplification of the target cDNA were designed to have an
annealing temperature around 60˚C. The forward primer (Gene-HF) for the target included
5′-CTGGTGCCGCGCGGCAGC-3′ (the 5' end of the adapter sequence) following the 20-bp
length of the 5' end of the target gene. The reverse primer (Gene-HR) for target gene amplifica-
tion included "GATGATGATTCCTCCTCC" the complementary sequence of the 3' end of the
adapter following 20-bp sequence reverse-complement to the 3' end of the target gene. Within
Gene-HR$_2$, a 20-bp sequence reverse-complement to a termination codon was removed at the
3' end of the target gene sequence to facilitate expression of the C-terminal tagged fusion pro-
tein. In conjunction with the rapid high-fidelity DNA polymerase, MCLAB (Beijing Tsingke
Biotech Co., Ltd., China), a forward primer "HighF" (5′-GGAGGAGGAATCATCATC-3′) was
designed at the 3' end of the adapter sequence. The reverse primer "HighR"
(GCTGCCGCGCGGCACCAG)" was at the 5' end of the adapter, and its reverse and complemen-
tary sequence was generated to linearize the pSDB1 to pSDB12 vectors via PCR amplification.
The Gene-HF and Gene-HR$_1$ primer pair were used to amplify cDNA fragments, which were
then cloned into N-terminal-tagged vectors, whereas Gene-HF and Gene-HR$_2$ primers were
used to clone cDNA fragments into the C-terminal-tagged vectors. For accurate quantification,
all DNA fragments were purified using the SanPrep Column PCR Product Purification Kit
(Shanghai Sangon Biotech Co., Ltd. China), following the manufacturer's instructions. There-
after, 0.03 pmol of insert and 0.015 pmol of vector (at a molar ratio of 2:1) were mixed in a
10 μL system, made up to volume with ddH$_2$O. The quality of a gene or vector can be esti-
mated using the following equation.

$$\text{Quality of insert} = 0.02 \times \text{No. (base pair) (Unit : ng)}$$

$$\text{Quality of vector} = 0.01 \times \text{No. (base pair) (Unit : ng)}$$

   The HTFC reaction mixture contained 0.015 pmol of linearized vector (the amount of
which can be calculated by multiplying the base pair number of vectors per nanogram by 0.01)
and an appropriate amount of insert DNA fragment (with an insert to vector molar ratio of 1:2
to 1:10), to which ddH$_2$O is added to a total volume of 10 μL. The reaction mixtures were
chemically transformed into 50 μL of chemically competent cells. The *recA*-defect cloning
strains, such as DH5α or Top10, were used to prepare chemically competent cells, unless indi-
cated otherwise. All 12 mixtures were then incubated for 30 min on ice. After heat shock at
42˚C for 45 s, 350 μL of LB medium was added to each of mixture. After 1 h shaking at 37˚C
and 350 rpm with a thermomixer, the content of each mixture was plated onto an LB agar
plate containing properly antibiotics (50 μg/mL ampicillin or 50 μg/mL kanamycin),

depending on the vectors used. The plates were then incubated at 37°C overnight. The next day, colonies from each construct were picked for PCR confirmation using 2×Utaq PCR Mas-terMix (Beijing Zoman Biotechnology Co., Ltd. China) and vector-specific primers (S4 Table). Each colony was inoculated in LB medium with antibiotics for overnight culture. Plasmids were prepared using the SanPrep Column Plasmid Mini-Preps Kit (Sangon Biotech Co., Ltd., China), following the manufacturer's instructions. After PCR confirmation, products were visualized using a UV transilluminator. All clones containing inserts of the expected size were finally confirmed by DNA sequencing at Tsingke Co., Ltd.

Genes from *Mycobacterium tuberculosis* strain ATCC 25618/H37Rv, namely, *Mqo* (Gene ID: 888544), *ThrS* (Gene ID: 888211), and *IleS* (Gene ID: 886412), with respective lengths of 1482, 2079, and 3216 bp, were used to evaluate cloning efficiency. The primers used are listed in S3 Table. Three colonies from the LB plates inoculated with each of the three transformants were randomly selected for positive recombinant colony screening based on colony PCR using sequencing primers (S4 and S5 Ts).

## Development of a multiple gene co-expression system based on the HTFC method

Using the HTFC method, we developed a multi-gene co-expression system using a polycis-tronic bacterial construct. All genes, except the first, were inserted in series with a 25-bp linker sequence (TTAACTTTAAGAAGGAGATATACAT) at the 5' end of the target gene, which con-tained a transcriptional enhancer and ribosomal binding site sequence (Showing in Fig 4). For the Trim6-L-Trim61-pSDB1 vector construct, Trim6 was initially inserted into the pSDB1 vec-tor according to the recommended procedure for HTFC. The transitional vector, Trim6-pSDB1, was prepared for the next round of construction. The forward primer HighF and reverse primer Trim6-pSDB1-VR (18 bases reverse-complementary to the 5' end of the linker sequence, followed by the complementary sequence of the 3' end of the *Trim6* gene) were used for linearized vector amplification. Primers Trim61-LF and Trim61-HR were designed to amplify the secondary target gene, *Trim61*. The forward primer contained the 3' end of the linker sequence (18 bases), followed by the 5' end of the Trim61 sequence. The reverse primer Trim61-HR includes "GATGATGATTCCTCCTCC" the complementary sequence of the 3' end of the adapter following 20-bp sequence reverse-complement to the 3' end of the target gene. In this way, the linearized Trim6-pSDB1 with an additional linker sequence at the 3' end and 18-bp linker sequence at the 5' end recombined with Trim61, which shared an identical adaptor at both the 5' and 3' ends. The recombinant plasmid, Trim6-L-Trim61- pSDB1, was identified by sequencing. Using the same strategy, the Trim6-L-Trim61- pSDB7 and ScEsa1-L-Epl1-L-Yng2-L-Eaf6-pSDB1 vectors were con-structed. To co-express a larger number of genes, a multi-thread simultaneous cloning scheme was developed for efficient construction of the co-expression vector. Lsm7-L-Lsm5-L-Lsm6-pSDB1 and Lsm8-L-Lsm4-L-Lsm3-L-Lsm2-pSDB1 vectors were con-structed, and a DNA fragment containing Lsm8-L-Lsm4-L-Lsm3-L-Lsm2 was inserted into the Lsm7-L-Lsm5-L-Lsm6-pSDB1 vector to obtain an Lsm7-L-Lsm5-L-Lsm6-L-Lsm8-L-Lsm4-L-Lsm3-L-Lsm2-pSDB1 co-expression vector (shown in S2 Fig).

## Small-scale recombinant protein expression analysis

All the recombinant expression vectors were transformed into *E. coli* BL21 (*DE3*). Single-col-ony cells were inoculated into 5 mL of LB medium at 37°C with shaking at 220 rpm until an $OD_{600}$ of 0.6–0.8 was reached, at which point, the cultures were induced with the addition of

isopropyl β-D-1-thiogalactopyranoside (IPTG) at a final concentration of 0.4 mM for 3 h at 37˚C. Following the induction of expression, 200 μL of the cell culture was withdrawn and centrifuged at 13,000 rpm for 2 min. After discarding the supernatant, the cells were resuspended in 20 μL of ddH$_2$O, to which 5 μL of 5× protein loading buffer was added. The samples were then boiled at 100˚C for 5 min and thereafter run on SDS-PAGE gels.

## Results

### General approach of HTFC

We constructed a bacterial expression system as an example based on 12 ubiquitous prokaryotic expression vectors (Table 1). These vectors can be divided into different categories based on their tags, affinity, solubilization, and utility. To obtain universal overhangs for each recombinant protein, we designed a thrombin site and tobacco etch virus (TEV) site between the fusion tag and the protein of interest to remove the N-terminal and C-terminal tags, respectively. Second, we created a 36-bp DNA adaptor, replacing the original multiple cloning site, which contains many restriction sites in each vector. Next, to linearize the 12 vectors, we designed a pair of primers "HighF and HighR" according to the sequence of the DNA adaptor. This strategy gives the user the ability to linearize all modified expression vectors by PCR amplification with one pair of primers. Using this method, the linearized vector and insert are amplified separately, which is beneficial for the large-scale preparation of vectors in the laboratory for cloning different target genes. Considering single gene cloning, the amplified cDNA ends with 18 bases overlapping with the ends of all linearized vectors. Mixtures of each insert and linearized vector were directly transformed into competent *E. coli* cells to obtain the desired clones (Fig 2).

To eliminate false-positive colonies harboring the template circular plasmid, we inserted a lethal *ccdB* gene expression core box [36] (containing the *Cat* promoter, the open reading frame of a chloromycetin-resistance gene, and the open reading frame of the *ccdB* gene) into 12 first-generation high-throughput prokaryotic expression vectors (pSDV1–V12). The *ccdB* protein-resistant *E. coli* strain DB3.1 (AngYu Biotechnologies, China) was used to clone the *ccdB* gene in pSDV1~V12, which contain a streptomycin-resistance gene, and we used combinations of three antibiotics to select for *ccdB* recombinant plasmids (either Str[+]Cm[+]Kan[+] for pFastB1~B6 or Str[+]Cm[+]Amp[+] for pSDB7~B12, as shown in S3 Fig). As secondary-generation

**Table 1. pFastB1~B12 original vector information.**

| Vector name | Original vector | Size | Manufacturer |
|---|---|---|---|
| pSDB1 | pET-28a | 5369bp | Novagen Cat. No. 69864–3 |
| pSDB2 | pET-28a | 5369bp | Novagen Cat. No. 69864–3 |
| pSDB3 | pET-28s | 5668bp | Modified from pET-28a by Dr. Jiahai Zhou |
| pSDB4 | pET-28s | 5668bp | Modified from pET-28a by Dr. Jiahai Zhou |
| pSDB5 | pET-28MBP | 6552bp | Modified from pET-28b by Brian A. Davies, Darren S. Carney and Bruce F. Horazdovsky |
| pSDB6 | pET-28MBP | 6552bp | |
| pSDB7 | pGEX6p-1 | 4984bp | GE Healthcare. Code No.27-4597-01 |
| pSDB8 | pGEX6p-1 | 4984bp | GE Healthcare. Code No.27-4597-01 |
| pSDB9 | pETDuet-1 | 5420bp | Novagen Cat. No.71146-3 |
| pSDB10 | pColdl | 4407bp | TAKARA BIO INC. Cat. No. 3360 |
| pSDB11 | pET-15b | 5700bp | Novagen Cat. No. 69661–3 |
| pSDB12 | pET-22b | 5493bp | Novagen Cat. No. 69744–3 |

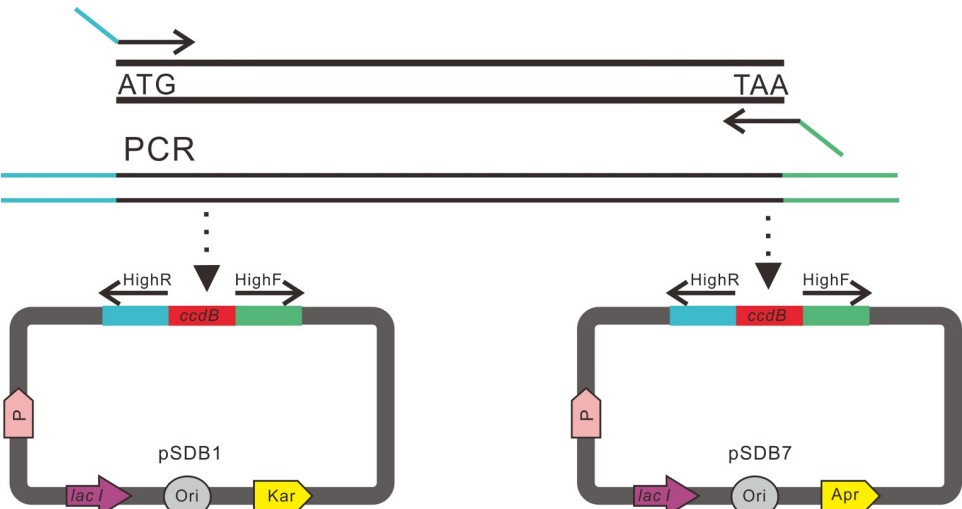

**Fig 2. A schematic diagram illustrating the high-throughput construct platform based on the high-throughput FastCloning method.** The second-generation high-throughput prokaryotic expression vectors share the same sequence elements, including a *ccdB* gene expression core box (red box), upstream box for primer HighR binding (blue box), and downstream box for primer HighF binding (green box). The same pair of HighF and HighR primers can linearize all 12 pSDB1–B12 vectors via PCR, whereas primers for amplifying insert fragments share a homologous overlap for the upstream or downstream box, which facilitates high-throughput cloning of a single gene in multiple vectors. The insert fragment was mixed with the linearized vector, and the recombinant plasmid was used to transform *Escherichia coli* DH5α competent cells. In the recombinant plasmid, the insert fragment replaced the ccdB box, thereby facilitating DH5α cell survival, whereas *ccdB* protein expression from non-recombinant plasmids was lethal to DH5α. P in the figure represents promoter. Apr: Ampicillin resistance gene. Kar: Kanamycin resistance gene.

high-throughput prokaryotic expression vectors (pSDB1–B12), the ccdB reading frame is replaced by the target DNA fragment, allowing cells to express the recombinant plasmid to propagate. Cells containing non-recombinant vectors still expressed *ccdB* and therefore leading to death. Instead of using *Dpn*I endonuclease to digest the template plasmid, we initially developed a high-efficiency DNA assembly platform independent of any biologically engineered enzymes, with the exception of high-fidelity DNA polymerase during PCR amplification.

To evaluate the high-throughput prokaryotic expression system that we constructed, three *M. tuberculosis* genes (*Mqo*, *ThrS*, and *IleS*) ranging in length from 1500 to 3000 bp were chosen to construct encoding fusion proteins. The amplified DNA fragments of these genes were mixed with 1 μL of linearized pSDB1–B12 vectors at a molar ratio of 2:1 and immediately transformed into DH5α competent cells. During the initial screening cycle, we randomly picked three colonies for colony PCR validation using appropriate sequencing primers. With high recombinant efficiency, we rapidly acquired at least nine different recombinant plasmids containing the same gene (12 for *Mqo*, 11 for *ThrS*, and 9 for *IleS*) (S4–S6 Figs). Following the second cycle of screening, we obtained 12 different recombinant plasmids for each of the three target genes (Fig 3A, 3C, and 3E). After obtaining these 12 recombinant plasmids, we subsequently used them to transform competent *E. coli* BL21 (*DE3*) cells to evaluate protein expression. The recombinant proteins encoded by all three genes were detected using SDS-PAGE (Fig 3B, 3D, and 3F). Given that we detected certain differences in protein expression levels, we modified the conditions of the expression system to optimize protein expression. These data revealed that high efficiency and attainable protein expression could be achieved using the HTFC method, which we envisage could be widely applied to optimize protein expression conditions for further studies in structural biology and synthetic biology.

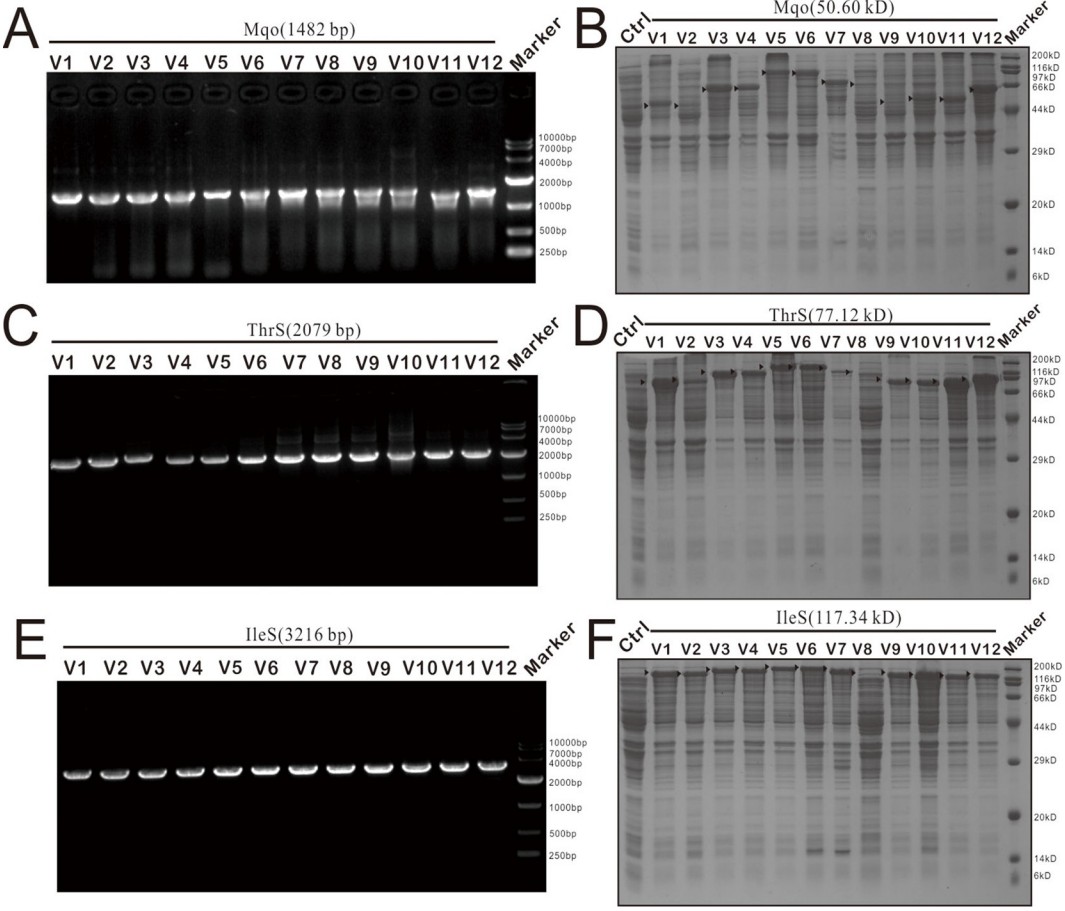

**Fig 3. Recombination validation and protein expression analysis for the high-throughput FastCloning method.** (A) Recombinant plasmids containing the *Mqo* gene were validated by PCR, using the same primers used to amplify the insert. Validation of the *ThrS* and *IleS* recombinant plasmids is shown in (C) and ©, respectively. (B) The *Mqo* recombinant plasmids were used to transform the *Escherichia coli* BL21 (DE3) strain, and recombinant protein expression was analyzed using SDS-PAGE. Fusion proteins of the expected size are denoted by black triangles. The analysis of ThrS and IleS protein expression is shown in (D) and (F), respectively.

### Construction polycistronic expression system based on the HTFC method

In prokaryotes, functionally related genes can be clustered together following a single promoter, and multiple protein products can be generated from a single messenger RNA strand called polycistron [37]. However, creating a polycistronic construct for bacterial expression is relatively complicated and time-consuming. Based on a preliminary experiment, we used secondary-generation high-throughput prokaryotic expression vectors (e.g., pSDB1 from the original pET-28a) to produce multiple recombinant proteins. To design a polycistronic system based on pSDB1, which contains the T7 promoter, to regulate the expression of the target gene, we generated an RNA transcript containing tandem translation cassettes (coding regions), each preceded by *E. coli* translation signals. Therefore, we designed a 25-bp linker sequence containing a transcriptional enhancer and a ribosomal binding site (RBS), which enables efficient translation of each coding region in the polycistronic transcript.

The construction of the pSDB1 polycistronic expression plasmid with four genes was aided by the modular design of each translation cassette. First, the user needs to determine the sequential position of each gene in the blueprint of polycistronic mRNA, especially the first

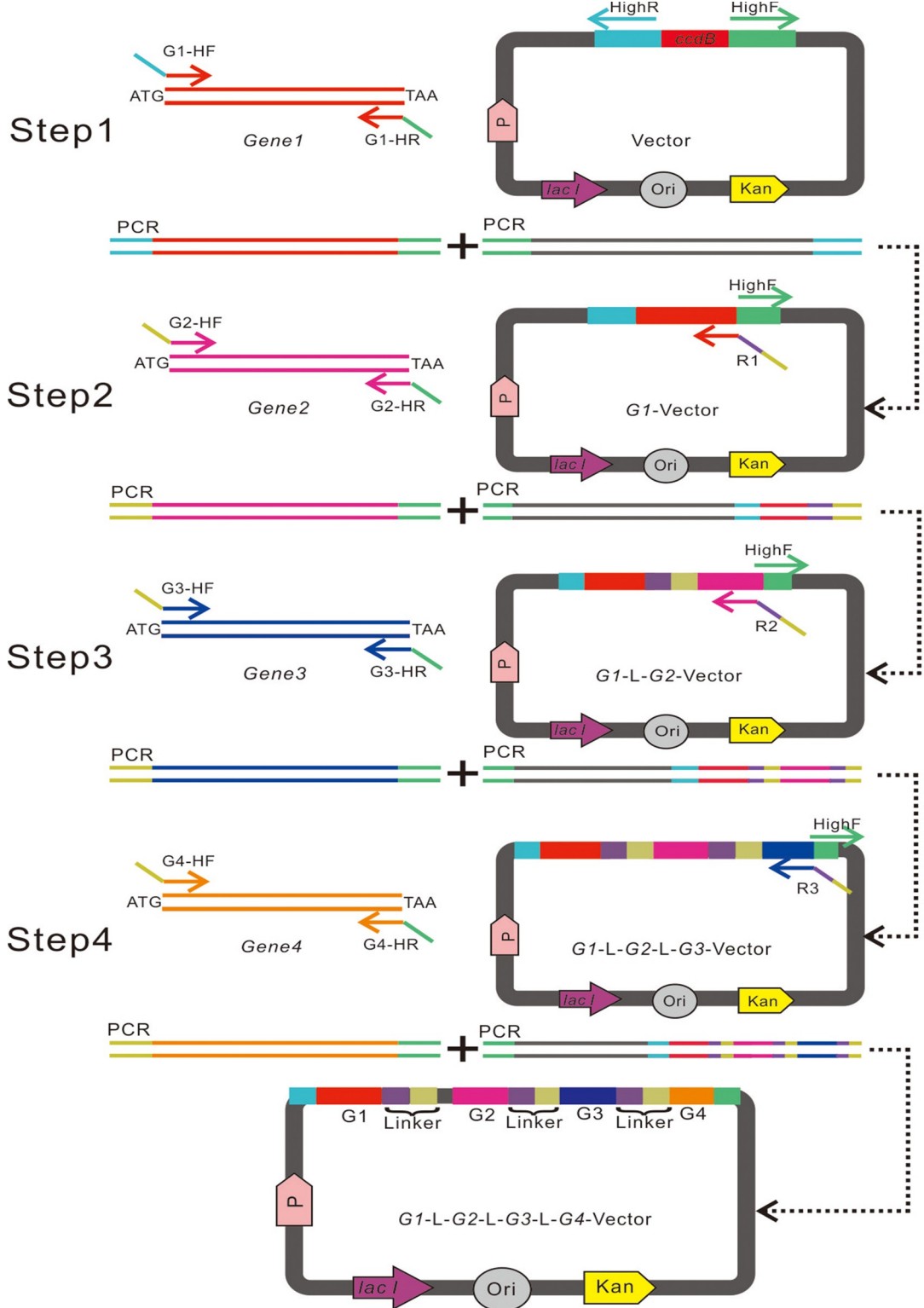

**Fig 4. A schematic diagram illustrating the multiple gene co-expression system based on the high-throughput FastCloning method.** Multiple gene co-expression systems were constructed using a stepwise approach. Step 1: Gene1 is inserted into a high-throughput vector. Step 2: The recombinant plasmid G1-Vector is linearized using the HighF and R1 primer pair. A 25-bp linker sequence is added to the linearized G1-vector fragment using the R1 primer. G2-HF shares a homologous 18-bp (denoted in yellow) overlapping with the R1 primer. Step 3: The G1-L-G2-Vector is linearized using the

HighF and R2 primer pair, followed by the insertion of Gene3. Step 4: The G1-L-G2L-G3-Vector is linearized using the HighF and R3 primer pair, followed by the insertion of Gene4. All homologous sequences are indicated by the same color.

gene. pSDB1 contains an original translation cassette composed of the *T7* gene and translation initiation signals, including the translational enhancer and the Shine-Dalgarno sequence, positioned before the cloning site for the first gene to be overexpressed. Therefore, the first gene (cDNA with start codon ATG) was added to pFastB1 using the regular method of HTFC. After the first gene was inserted into the vector, each subsequent translation cassette was sequentially inserted into pSDB1. Each translation cassette contained the same 5' end linker region composed of a translational start signal and a different gene (cDNA with start codon ATG) located on the 3' end. During each vector construction cycle, one translation cassette was added downstream of the last translation cassette. The linker sequence was added to the linearized vector using a reverse primer containing 20 bases 3' end cDNA of the last gene, followed by the 25-bp linker, and its reverse and complementary sequence was generated to linearize the vector. The forward primer (HighF) for vector amplification was always used in the linearization process for all vectors. Based on the principles of primer design to amplify the inserts for HTFC, all candidate inserts shared the same ends because the primers for gene amplification inserted a specific sequence (dependent on the gene sequence) and an additional 18 bases 5′-GATGAT GATTCCTCCTCC-3′ overlapping with the vector ends. Therefore, the user can randomly select any candidate gene as the next translational cassette without any limitations. When the number of co-expressed genes is greater than three, the user can generate a multi-thread simultaneous cloning scheme, which enables the recombination co-expression vector to be obtained in a relatively short time (see S2 Fig). This strategy affords a flexible arrangement of polycistronic mRNA without extra primers or entry vectors (Fig 4).

Using this co-expression system, we successfully expressed Trim6-Trim61 tRNA methyltransferase dual protein, NuA4 acetyltransferase tetrad protein, and Lsm7-5-6-8-4-3-2 snRNP heptameric protein complexes. For the expression of Trim6-Trim61- pFastB1, the Trim6 protein fused to a 6×His tag was eluted from $Ni^{2+}$-immobilized metal affinity chromatography using imidazole, whereas the untagged protein Trim61 formed a tight protein complex with Trim6, which could be eluted with imidazole buffer (Fig 5A). For the expression of Trim6-Trim61-pSDB7, the Trim6 protein was fused to the GST tag, while Trim61 did not fuse to a tag but formed a complex with Trim6 (Fig 5B). In the case of Esa1-Epl1-Yng2-Eaf6-pSDB1 expression, the 6×His-tagged Esa1 protein co-eluted with other NuA4 acetyltransferase core complex components, namely, Epl1, Yng2, or Eaf6. We also detected an approximately 43 kDa contaminant protein that co-eluted with the NuA4 acetyltransferase core complex, which was identified as the *E. coli* endogenous protein EF-Tu, as previously reported (Fig 5C) [38–40]. For Lsm7-5-6-8-4-3-2-pSDB1 expression, the 6×His-tagged Lsm7 was co-eluted with the other U6 snRNPs of the heptameric protein complex (Fig 5D). The snRNP components had similar molecular weights, and the composition of the eluted complex was further identified using mass spectrometry.

## Discussion

The high-throughput FastCloning method is highly efficient and cost effective, and is the first cloning strategy involving simultaneous cloning whereby a gene is simultaneously inserted into multiple vectors. This new strategy eliminates the need for specific primers for each unique vector. Compared with the In Fusion method, there is no need for *Dpn1* digestion and incubation *in vitro* (Fig 6), nor the use of the commercial the In-Fusion HD Cloning Kit

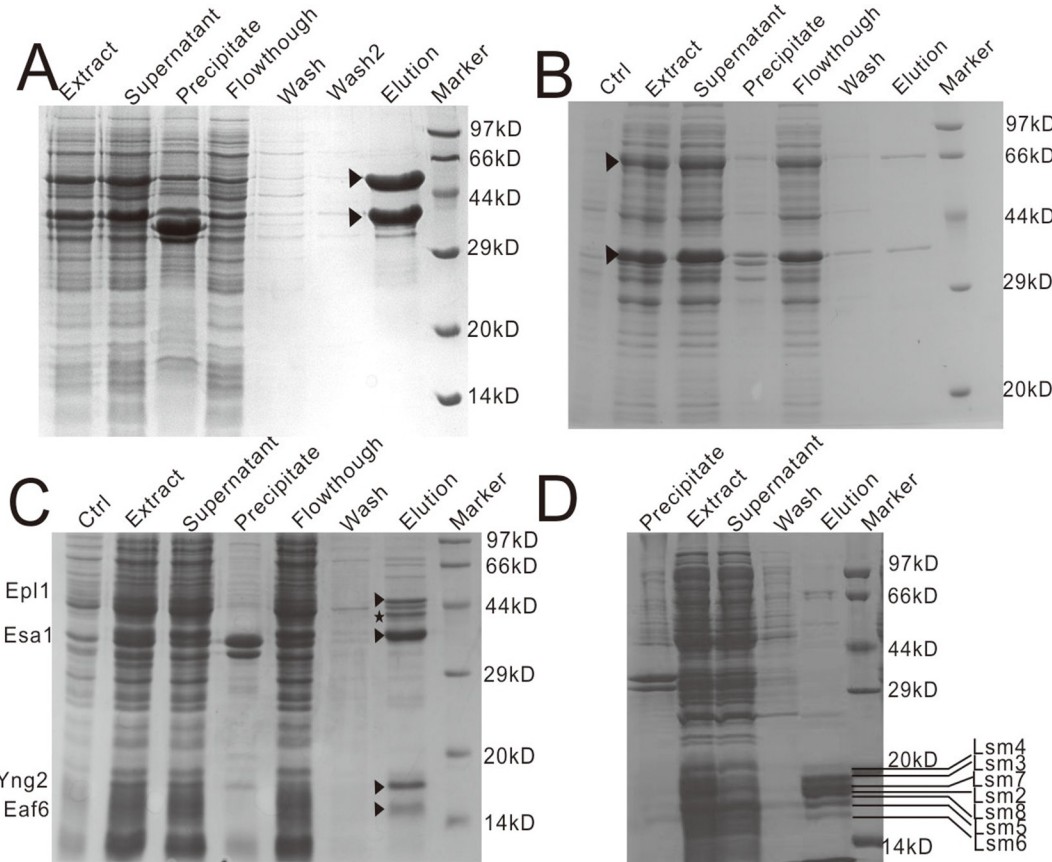

**Fig 5. Co-expression and co-purification of protein complexes based on the multiple gene co-expression system.** (A) Co-expression of the Trim6-Trim61 complex with two subunits of the Trim6-Trim61-pSDB1 vector. The Trim6 protein fused to a 6×His tag was harvested using Ni-NTA resin, whereas the untagged Trim6 protein bound tightly with the Trim6 protein forming a complex, which was eluted with imidazole buffer. (B) Co-expression of the Trim6-Trim61 complex in the Trim6-Trim61-pSDB7 vector. Trim61 did not fuse to a tag but formed a complex with Trim6 fused to a GST tag. The complex was purified by glutathione affinity chromatography. (C) Co-expression of the NuA4 complex comprising four subunits. Eas1 was expressed by fusing to a 6×His tag, and formed the NuA4 complex with the Epl1, Yng2, and Eaf6 subunits. This complex was then eluted with imidazole buffer. A contaminant protein is denoted by a black star. (D) Co-expression of the Lsm7-5-6-8-4-3-2 snRNPs complex comprising seven subunits in the pSDB1 vector. Lsm7 fused to a 6×His tag was co-eluted with other U6 snRNPs in a heptad protein complex. All target proteins are indicated by black triangles or labeled with the respective protein names.

(Takara, Japan). Thus, the HTFC method can be exploited for high-throughput DNA assembly, facilitating simultaneous cloning of a single gene into multiple vectors. More detailed information regarding the HTFC method compared to cloning using molecular cloning kits, such as Gibson Assembly® Master Mix (NEB, USA) and Trelief™ SoSoo Cloning Kit (Tsingke, China), is presented in S6 Table. In contrast, with the exception of DNA polymerase, there is no expenditure on bioengineering enzymes when using HTFC, and the cloning procedure is less complex, which represents an absolute advantage in terms of experimental costs (Fig 6).

In addition, the HTFC method can be applied in conjunction with multiple gene co-expression systems, independent of any specialized vectors, and facilitates the cloning of multiple genes into a single vector in a step-by-step manner or in a multi-thread simultaneous cloning scheme. The multiple-gene co-expression system based on the HTFC method utilizes polycistronic mRNA under the control of the same promoter. After the first gene, each translation cassette contains a 25-bp linker sequence, including a transcriptional enhancer and ribosomal

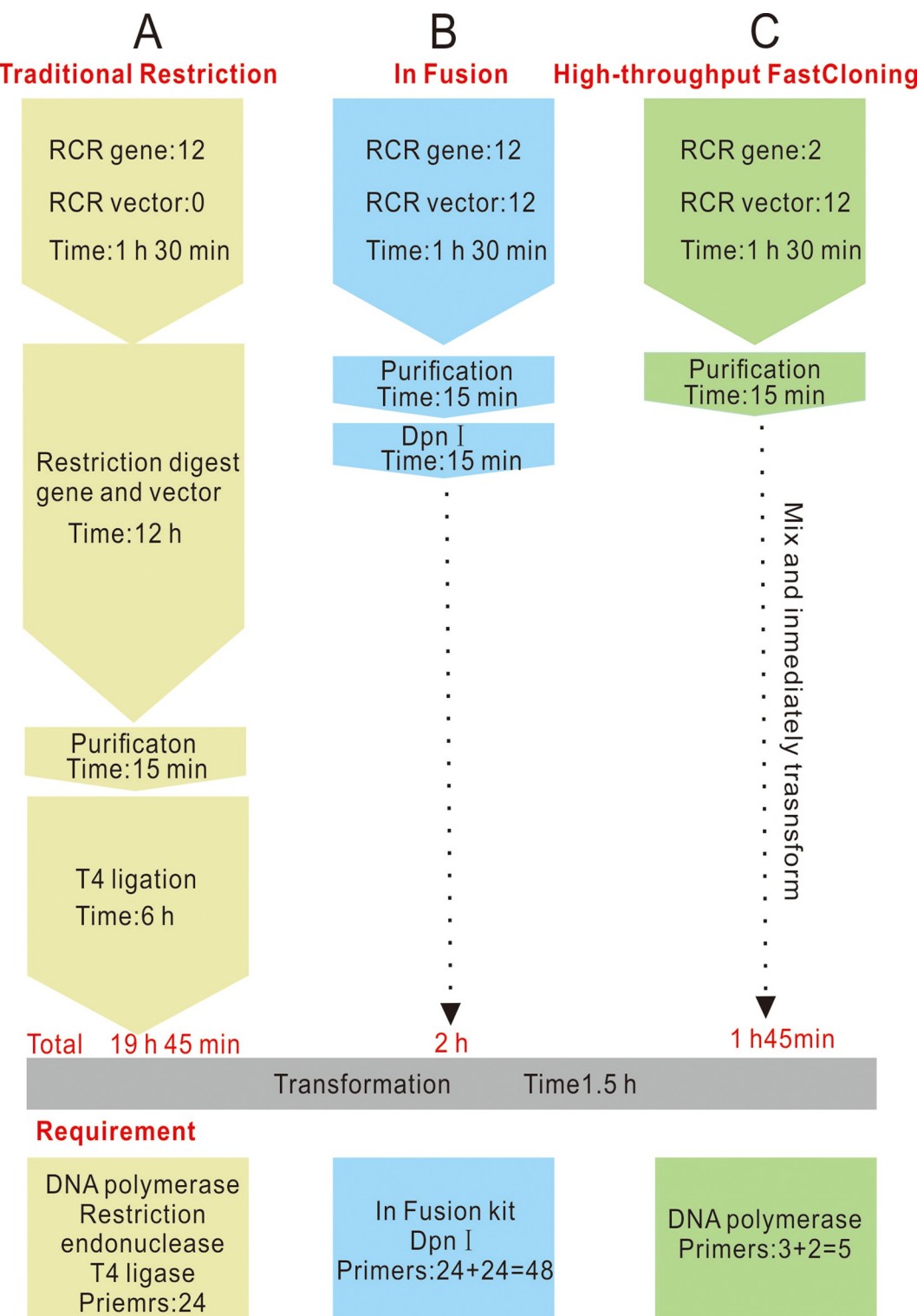

**Fig 6. A comparison of high-throughput FastCloning (HTFC) with current optimal protocols for different cloning procedure.** (A) The experimental workflow for traditional restriction cloning procedure. (B) The experimental workflow for in-fusion cloning procedure. (C) The experimental workflow for high-throughput FastCloning procedure.

binding site at the 5' end of the target gene. Simultaneously, an adapter sequence was added to the 3' end of the translation cassette. Therefore, each translation cassette can be randomly inserted into a polycistronic expression vector without sequence limitations. Theoretically, one can get any combination of multiple translation cassettes can be obtained in parallel without considering the cost.

## Supporting information

**S1 Fig. A diagram showing modification of a first-generation high-throughput vector.**
(TIF)

**S2 Fig. A diagram showing a multi-thread simultaneous cloning scheme for construction of a co-expression vector.**
(TIF)

**S3 Fig. A diagram showing ccdB core box cloning and screening.**
(TIF)

**S4 Fig. Colony PCR for the detection of recombinant plasmids harboring the Mqo gene.**
(TIF)

**S5 Fig. Colony PCR for the detection of recombinant plasmids harboring the ThrS gene.**
(TIF)

**S6 Fig. Colony PCR for the detection of recombinant plasmids harboring the IleS gene.**
(TIF)

**S1 Table. The information of second generation high-throughput prokaryotic expression vectors.**
(DOCX)

**S2 Table. Primers used to modify high-throughput constructs.**
(DOCX)

**S3 Table. Primers used to test the recombination efficiencies.**
(DOCX)

**S4 Table. Sequencing primers to high-throughput vector.**
(DOCX)

**S5 Table. The sequence of sequencing primers.**
(DOCX)

**S6 Table. Comparison of molecular cloning kit at home and abroad.**
(DOCX)

**S1 Data.**
(ZIP)

## Acknowledgments

We gratefully acknowledge the core facility staff members of the West China Hospital. We also thank Dr. Guangwen Lu, Dr. Rundong Zhang, and Dr. Junhong Han for providing plasmids and research guidance.

## Author Contributions

**Conceptualization:** Dan Su.

**Data curation:** Hua Jiang, Deren Lu, Yuejun Chen, Jun Chen, Xi Zhang.

**Formal analysis:** Yanjuan Chen.

**Funding acquisition:** Cheng Chen, Dan Su.

**Investigation:** Hua Jiang, Fan Meng, Guilin Luo, Yuejun Chen.

**Methodology:** Hua Jiang, Fan Meng, Deren Lu, Guilin Luo.

**Project administration:** Dan Su.

**Resources:** Dan Su.

**Validation:** Fan Meng.

**Visualization:** Yanjuan Chen.

**Writing – original draft:** Hua Jiang.

**Writing – review & editing:** Cheng Chen, Dan Su.

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
