## [Decision Letter · Decision Letter 0]

12 Jun 2022

PONE-D-22-13103High-throughput FastCloning technology: A low-cost method for parallel cloningPLOS ONE

Dear Dr. Su,

Thank you for submitting your manuscript to PLOS ONE. After careful consideration, we feel that it has merit but does not fully meet PLOS ONE’s publication criteria as it currently stands. Therefore, we invite you to submit a revised version of the manuscript that addresses the points raised during the review process.

We look forward to receiving your revised manuscript.

Kind regards,

Paulo Lee Ho, Ph.D.

Academic Editor

PLOS ONE

Journal Requirements:

[This work was supported by grants from the National Key Research and Development Program of China (2017YFA0505903), the National Natural Science Foundation of China (31370735, 31670737 to D. S., 81671494 to C. C.), the Technology Department of Tianjin Foundation (19YFZCSN00470), 1·3·5 projects for disciplines of excellence, West China Hospital, Sichuan University (ZYJC18033), the Special Research Fund on COVID-19 of Sichuan Province (2020YFS0010), and the Key Project on COVID-19 of West China Hospital, Sichuan University (HX-2019-nCoV-044).]

 [This work was supported by grants from the National Key Research and Development Program of China (2017YFA0505903), the National Natural Science Foundation of China (31370735, 31670737 to D. S., 81671494 to C. C.), the Technology Department of Tianjin Foundation (19YFZCSN00470), 1·3·5 projects for disciplines of excellence, West China Hospital, Sichuan University (ZYJC18033), the Special Research Fund on COVID-19 of Sichuan Province (2020YFS0010), and the Key Project on COVID-19 of West China Hospital, Sichuan University (HX-2019-nCoV-044).]

Reviewers' comments:

Reviewer's Responses to Questions

**Comments to the Author**

1. Is the manuscript technically sound, and do the data support the conclusions?

Reviewer #1: Yes

Reviewer #2: Yes

2. Has the statistical analysis been performed appropriately and rigorously? 

Reviewer #1: I Don't Know

Reviewer #2: N/A

3. Have the authors made all data underlying the findings in their manuscript fully available?

Reviewer #1: Yes

Reviewer #2: Yes

4. Is the manuscript presented in an intelligible fashion and written in standard English?

Reviewer #1: Yes

Reviewer #2: Yes

5. Review Comments to the Author

Reviewer #1: The authors describe a fast-cloning method that had been published previously (FastCloning: a highly simplified, purification- free, sequence- and ligation-independent PCR cloning method, 2011), therefore, the technique is interesting but not new, once it was already published. However, in this work, this method was further explored to construct different vectors of expression, and three genes from Mycobacterium tuberculosis were used to evaluate the cloning efficiency of molecular cloning techniques, therefore, extending the previous studies on this technique which may be interesting to be explored for the scientific community.

Some questions/comments

1- In Materials and methods, lanes 211-220 say that the amplicon was mixed with an amplified vector. However, since in this case DpnI was not added, and no overhang was created, should some denaturing event carried out to help to promote the base pairing?

2- Fig. 2- I think this figure can be simplified by just adding two vectors since apart from the resistance gene, they are all the same. Maybe two vectors and inside one (pFastB1-6) and the other (pFastB7-12).

3- Fig. 5. Is this gel performed with reduction conditions? Because is not clear why there are 2 bands in the elution, could you explain this a little better?

Reviewer #2: The article is well written. Despite not being scientifically original, it brings an elegant and alternative method of cloning in expression vectors. The constructs were analyzed by restriction pattern, but I did not find data that they were sequenced, if any mutation occurs, especially in polycistronics, it can harm the entire genetic construct. It is also not clear what analysis was done on the primers, what are the chances of self annealing, for example?

In Figure 3, it would be interesting to place the western blot and/or the negative controls to compare the expression of the proteins. The same goes for figure 5.

In figure 2, describe P, apr and Kar. The way the arrow indicates the target for P suggests that the gene would be inserted in this region.

The difference between pFASTV1-V12 and pFASTB1-B12 is unclear.

In table 1 you could add the size in Kpb of the vectors used.

I suggest changing the name of the vectors, as pFAST already exists, they are vectors marketed by Thermofisher for the baculovirus system.

6. PLOS authors have the option to publish the peer review history of their article (what does this mean?). If published, this will include your full peer review and any attached files.

Reviewer #1: No

Reviewer #2: No

---

## [Author Response · Author response to Decision Letter 0]

30 Jul 2022

Dear Editor Paulo Lee Ho, Dear reviewers

Thank you for your letter. We were pleased to know that our work was rated as potentially acceptable for publication in Journal POLS ONE, subject to adequate revision. We thank the reviewers for the time and effort that they have put into reviewing the previous version of the manuscript. Their suggestions have enabled us to improve our work. Based on the instructions provided in your letter, we uploaded the file of the revised manuscript. Accordingly, we have uploaded a copy of the original manuscript with all the changes highlighted by using the track changes mode in MS Word. Appended to this letter is our point-by-point response to the comments raised by the reviewers. The comments are reproduced and our responses are given directly afterward in a different color (red). We would like also to thank you for allowing us to resubmit a revised copy of the manuscript.

Reviewer #1: The authors describe a fast-cloning method that had been published previously (FastCloning: a highly simplified, purification- free, sequence- and ligation-independent PCR cloning method, 2011), therefore, the technique is interesting but not new, once it was already published. However, in this work, this method was further explored to construct different vectors of expression, and three genes from Mycobacterium tuberculosis were used to evaluate the cloning efficiency of molecular cloning techniques, therefore, extending the previous studies on this technique which may be interesting to be explored for the scientific community.

Some questions/comments

1- In Materials and methods, lanes 211-220 say that the amplicon was mixed with an amplified vector. However, since in this case DpnI was not added, and no overhang was created, should some denaturing event carried out to help to promote the base pairing?

Answer: Thanks for your question. We use DpnI is to digest the methylated parent DNA templates (if in a plasmid) for PCR amplification, avoid causing false positives after transformation. The lethal gene (ccdB ) was added into our modified vector at the insert region. A successful insertion will completely replace ccdB with the investigator’s insert of interest. Hence correct clones are identified much more efficiently, as those that do not contain the desired insert should not grow. Therefore, it is not necessary to add the dpn1 digestion template. As for how overhang formed, although the detailed mechanism is not known, it is likely that the 3’ exonuclease activity of the high-fidelity DNA polymerase directly creates sticky ends for the overhang of the vector and insert, allowing them to form a circular construct with nicks.

2- Fig. 2- I think this figure can be simplified by just adding two vectors since apart from the resistance gene, they are all the same. Maybe two vectors and inside one (pFastB1-6) and the other (pFastB7-12).

Answer: Thanks for your suggestion, we have made correction according to your comments in figure 2.

3- Fig. 5. Is this gel performed with reduction conditions? Because is not clear why there are 2 bands in the elution, could you explain this a little better?

Answer: Thanks for your question. Here we make further explanation, in the polycistronic expression system, as the schematic diagram illustrating in figure 4, we generated the co-expression plasmid, Trim6-Trim61- pFastB1, the Trim6 protein fused to a 6×His tag , whereas the untagged protein Trim61 has interaction with Trim6,they can form stable complex, so there are two bands in elution (Fig.5A). By the same token, Epl1,Yng2,Eaf6 can form complex with Esa1, so there are 4 bands in elution(Fig.5C).

Reviewer #2: The article is well written. Despite not being scientifically original, it brings an elegant and alternative method of cloning in expression vectors. The constructs were analyzed by restriction pattern, but I did not find data that they were sequenced, if any mutation occurs, especially in polycistronics, it can harm the entire genetic construct. It is also not clear what analysis was done on the primers, what are the chances of self annealing, for example?

Answer：Thanks for your question. We used the DNA polymerase contained in MCLAB is Goldenstar T6 DNA polymerase modified by genetic engineering. It has a very high fidelity (60 times of Taq), not easy to produce mutations during PCR. And we sequenced the whole vectors (pSDB1-B12) to ensure the correctness of the vectors’ base sequence. The sequenced data show in supplementary information folder named sequencing data. But we can’t guarantee that mutations won’t be introduced into the construct for other reasons. Considering the cost of sequencing, we only sequenced the insert gene of the three genes from Mycobacterium tuberculosis were used to evaluate the cloning efficiency of molecular cloning techniques, and the constructed plasmid was transferred to BL21(DE3) for protein expression. The results showed that the molecular weight of the expressed protein conformed to the theoretical molecular weight. It is proved that the gene expression element on the constructed plasmid is complete. As for the analysis on the primers, we calculated the Tm value of primer with software Primer 5, optimized the Tm values of high F and high R for vector amplification, and tested that the optimal annealing temperature is 60 ℃ by PCR experiments, won’t cause primer self annealing. The Tm value of primers used for gene amplification will change with different insert gene sequence. The annealing temperature should be adjusted according to specific conditions. Generally, it is appropriate to subtract 5 ℃ from Tm value.

In Figure 3, it would be interesting to place the western blot and/or the negative controls to compare the expression of the proteins. The same goes for figure 5.

Answer: We appreciate the reviewer for this kind recommendation. In figure 3A,3D and 3F, the negative controls are shown in the first lane, the expression of the proteins are denoted by black triangles according to the expected size. In the figure, the protein expression bands of some vectors are obvious, such as V1, V3, V4, V5, V6, V7, V11and V12 of Mqo. We admit that we cannot guarantee all 12 expression plasmids in the high-throughput plasmid construction system can express the target protein. Our purpose is to screen the best expression vector of the target protein by High-throughput Fast-Cloning technology. We also appreciate your proposal to compare protein expression with the negative controls using Western blot. However, due to the impact of the COVID-19 in Chengdu, we are in control area, resulting in the inability to purchase the corresponding antibodies for WB experiment. We sincerely hope you can understand. As for figure 5, the expressed protein Trim6 (Fig.5A),Eas1(Fig.5C),Lsm7(Fig.5D) fused to a 6×His tag, this tag will binding on Ni-NTA resin, and the target protein can be eluted with high concentration of imidazole, the bands in elution indicates that the target protein is expressed. In figure 5B, GST tag was fused to Trim6, the target protein complex was purified by glutathione affinity chromatography, eluted by reduced glutathione.

In figure 2, describe P, apr and Kar. The way the arrow indicates the target for P suggests that the gene would be inserted in this region.

Answer: Thanks for your suggestion. P: promoter. Apr: Ampicillin resistance gene. Kar: Kanamycin resistance gene. We revised the figure 2 to make it clearer that the gene will insert between high F and high R.

The difference between pFASTV1-V12 and pFASTB1-B12 is unclear.

Answer: We thank the reviewer for raising this question. We created a 36-bp DNA adaptor, replacing the original multiple cloning site, which contains many restriction sites in each vector(pFASTV1-V12), to linearize the 12 vectors, we designed a pair of primers “HighF and HighR” according to the sequence of the DNA adaptor. This strategy gives the user the ability to linearize all modified expression vectors by PCR amplification with one pair of primers. For pFASTB1-B12, a lethal ccdB gene expression core box (containing the Cat promoter, the open reading frame of a chloromycetin-resistance gene, and the open reading frame of the ccdB gene) was inserted into pFASTV1-V12 between HighF and HighR, which can eliminate false-positive colonies harboring the template circular plasmid without using DpnI digested the template plasmid.

In table 1 you could add the size in Kpb of the vectors used.

Thanks for your suggestion, we have added the size of the vectors in table 1.

I suggest changing the name of the vectors, as pFAST already exists, they are vectors marketed by Thermofisher for the baculovirus system.

Answer: Thank you for your kind reminder. We have changed the name of the vectors pFAST to pSD in the revised manuscript.

---

## [Decision Letter · Decision Letter 1]

17 Aug 2022

High-throughput FastCloning technology: A low-cost method for parallel cloning

PONE-D-22-13103R1

Dear Dr. Su,

We’re pleased to inform you that your manuscript has been judged scientifically suitable for publication and will be formally accepted for publication once it meets all outstanding technical requirements.

Kind regards,

Paulo Lee Ho, Ph.D.

Academic Editor

PLOS ONE

Additional Editor Comments (optional):

Reviewers' comments:

Reviewer's Responses to Questions

**Comments to the Author**

1. If the authors have adequately addressed your comments raised in a previous round of review and you feel that this manuscript is now acceptable for publication, you may indicate that here to bypass the “Comments to the Author” section, enter your conflict of interest statement in the “Confidential to Editor” section, and submit your "Accept" recommendation.

Reviewer #1: All comments have been addressed

2. Is the manuscript technically sound, and do the data support the conclusions?

Reviewer #1: Yes

3. Has the statistical analysis been performed appropriately and rigorously? 

Reviewer #1: I Don't Know

4. Have the authors made all data underlying the findings in their manuscript fully available?

Reviewer #1: Yes

5. Is the manuscript presented in an intelligible fashion and written in standard English?

Reviewer #1: Yes

6. Review Comments to the Author

Reviewer #1: All questions raised in the previous review was properly addressed and I feel that this manuscript is now acceptable for publication.

7. PLOS authors have the option to publish the peer review history of their article (what does this mean?). If published, this will include your full peer review and any attached files.

Reviewer #1: No

---

## [Editor Report · Acceptance letter]

30 Aug 2022

PONE-D-22-13103R1 

High-throughput FastCloning technology: A low-cost method for parallel cloning 

Dear Dr. Su:

I'm pleased to inform you that your manuscript has been deemed suitable for publication in PLOS ONE. Congratulations! Your manuscript is now with our production department. 

Kind regards, 

on behalf of

Dr. Paulo Lee Ho 

Academic Editor

PLOS ONE